# The State of Health and the Quality of Life in Women Suffering from Endometriosis

**DOI:** 10.3390/jcm11072059

**Published:** 2022-04-06

**Authors:** Monika Ruszała, Dominik Franciszek Dłuski, Izabela Winkler, Jan Kotarski, Tomasz Rechberger, Marek Gogacz

**Affiliations:** 1Chair and Department of Obstetrics and Perinatology, Medical University of Lublin, 20-090 Lublin, Poland; 2II Department of Gynecology, St John’s Center Oncology, 20-090 Lublin, Poland; ikochans@interia.pl; 3I Chair and Department of Oncological Gynecology and Gynecology, Medical University of Lublin, 20-093 Lublin, Poland; jan.kotarski.gabinet@gmail.com; 4II Chair and Department of Gynecology, Medical University of Lublin, 20-954 Lublin, Poland; rechbergt@yahoo.com (T.R.); gogacz@yahoo.com (M.G.)

**Keywords:** mental health, sexual dysfunction, sexuality, clinical management

## Abstract

Quality of life is related to good health, family relations, feeling of self-esteem, and ability to cope with difficult situations. Endometriosis is a chronic condition which affects different areas of life. The lack of satisfaction in everyday life is mainly due to constant pain. The process of adjusting to a life with illness is associated with negative emotions. The aim of the article is to review the current state of knowledge concerning the impact of social and medical factors on a population of women affected by endometriosis. Women with endometriosis have an impaired quality of life compared to the general female population. Psychological consequences of endometriosis include: depression, anxiety, powerlessness, guilt, self-directed violence, and deterioration of interpersonal relations. It may contribute to lower productivity at work and less satisfying intimate life. A multi-disciplinary, evidence-based care is needed. The disease can take away the ability to be physically active, obtain an education, work continuously, and interact with friends. Social support and cognitive-behavioral therapy are extremely important for healing.

## 1. Introduction

Endometriosis is a long-term, inflammatory, incurable disease usually occurring in young women of the reproductive age, which involves localization of the endometrium outside the uterine cavity [1,2]. The most frequent site of the changes is the small pelvis, however, there are cases when endometrial cells are present in distant organs, such as the gastrointestinal system, lungs, and brain [3]. The incidence of the disease is estimated at 10–15% (176 million women worldwide) [4]. Approximately 50% of infertile women suffer from endometriosis [5,6]. The causes of endometriosis are not fully understood. There are many theories concerning the origins of the disease. Immune processes certainly play a significant role [7]. Endometriosis affects women at the reproductive age as well as adolescents and very young girls. The disease is more and more often diagnosed in menopausal women [8]. The incidence of the disease is observed in every social and ethnic group. The course of endometriosis can be asymptomatic [9]. One of the biggest problems in women with the disease is late diagnosis. Due to atypical symptoms or a wide range of symptoms of pain and the lack of one accepted method or examination to detect endometriosis, patients seek help from doctors of various specialties for many months or even years—from a general practitioner through a urologist, surgeon, gastroenterologist, and finally a gynecologist [10].

Pain with the neuropathic or muscular components is the main symptom of the disease; depending on the location of an endometriosis focus, patients complain of pain within the reproductive system, urinary system, or lower part of the digestive system [11,12,13]. One fourth of the patients do not mention pain as a symptom. Difficulty in diagnosing the disease is also related to atypical symptoms, which can mask the nature of the disease. They include, among others: pain in the lumbar-sacral region of the spine, diarrhea, constipation, fatigue, flatulence, nausea, anxiety, depression, and headache [14]. Typical symptoms in patients with endometriosis are: abdominal pain, painful periods, dyspareunia, dysuria, pain during defecation, back pain, pain in the region of the urinary bladder, intestinal pain, bleeding from the area of the anus and urethra, and, finally infertility [15]. Pain is associated with menstrual bleeding—it appears before and during menses, it is located in the region of the pelvis minor, and it concerns the organs adjacent to the reproductive organ. Endometriosis is a chronic disease and leads to numerous complications [16]. Firstly, it stimulates the immune system, which is related to the occurrence of other diseases with the immune background (e.g., thyroid diseases, systemic lupus, joint inflammation) [17]. Secondly, the long-lasting disease leads to complications in the form of ovarian cysts formation, intraperitoneal adhesions (local inflammatory condition, infiltration of neighbouring tissues, collection of blood), infertility, and miscarriage [18]. All these symptoms, delayed diagnostics, and a long course of the disease lead to a significant decrease in the comfort of life of patients with endometriosis [19].

## 2. Quality of Life

The term “quality of life” appeared in medical literature in the 1960s and has become increasingly popular in recent decades. It is often defined as sense of well-being, health, comfort, and happiness experienced by an individual or group. Basic indicators include wealth, employment, the environment, physical and mental health, education, recreation and leisure time, social belonging, religious beliefs, safety, security, and freedom. Although happiness is subjective and difficult to measure, the rest of the parameters may be estimated [20]. In health care, quality of life is often regarded in terms of how a certain disease/state affects a patient on an individual level [21]. There are various questionnaires and tests examining the quality of life in patients with endometriosis. The most frequently used include: EHP-30 (Endometriosis Health Profile-30 items), EHP-5 (Endometriosis Health Profile-5 items), SF-36 (Short Form-36), WHO QOF-100 (World Health Organization Quality of Life Questionnaire), and WHO QOL BREF (World Health Organization Quality of Life Assessment) [22].

One of the most cited components in recent literature referred to endometriosis and how it may affect the quality of women’s lives are social support, self-image, relationships, profession, access to treatment, and infertility. In the study performed by Muharam et al. (2022), scientists noticed that the lowest scores referred to emotional disturbances in relationship between mothers and their children. They also stressed the possibility of alcohol addiction occurrence, attention deficit, or hyperactivity disorders. Another factor correlated with the disease and quality of life was available treatment method, which might enable a control of powerlessness in a daily activity. The analysis also showed moderate correlation between the disease, self-imagine, and social support [23]. In the study conducted by Bień et al. (2020), acceptance of illness in patients who took part in the research has maintained on moderate level. A total of 9 out of 10 women assessed the treatment as expensive. Better scores in well-being were achieved by women with high education. In the questionnaire, patients highlighted general health and local environment as important factors that had an impact on quality of life. Women with endometriosis may manifest lower quality of life in comparison to healthy women with the same symptoms. An interesting topic seems to be how age affects the perception of illness and well-being. Bień et al. described that older patients indicated worsen quality of life. Morandi et al. expanded this issue and noticed similarities and differences on different age. Common aspects in all patients with endometriosis concerned marital relations, social life, illnesses, and mental health. For women between 25 and 34 years of age, the most important aspects that affected well-being were employment, financial situation for those who were above 35, and education below 24. The next aspect concerned perception of one’s own body. The scientists after the analysis of the work concluded that women with obesity scored the lowest points. They were less satisfied with themselves and their intimate life [24,25].

## 3. Physical and Mental Symptoms of Endometriosis

The quality of life of women suffering from endometriosis is influenced by many factors. The most significant is unpredictable pain, occurring periodically, with different degrees of severity, which causes about 38% reduction of work productivity [26,27,28]. Several working days missed generate direct and indirect costs comparable to diabetes, migraine, asthma, or rheumatoid arthritis [29,30]. Pokrzywinski et al. (2020) report in their study that employed women who suffer from endometriosis may be 16.5 (±11.4) h unable to work per week, whereas the household group were 8.3 (±8.7) h per week [31]. They reported negative career effects or low performance related to endometriosis. In many cases, this is not only abdominal pain but also pain during defecation and urination, making daily life difficult. It is associated with constant taking of analgesics and a feeling of living with a chronic disease [32,33,34] (Figure 1).

Some women may additionally feel fear of pain occurrence, strong need to control it, intrusive worry thoughts, catastrophizing, self-blame, and ruminations [35] (Figure 2). Women with confirmed endometriosis experience significantly more symptoms of depression (standardized mean difference [SMD] of 0.71 (95% confidence interval [CI] 0.36–1.06)) and anxiety (SMD 0.60 (95% CI 0.35–0.84)) in comparison to healthy ones (SMD −0.01 (95% CI −0.17 to 0.15) for depression and SMD −0.02 (95% CI −0.22 to 0.18) for anxiety) [36]. Ceran et al. (2020) obtained comparable results in which a significant difference was found between the endometriosis and non-endometriosis groups regarding depression scores [37]. Autoimmune diseases, accompanying endometriosis, aggravate the somatic and mental condition of patients [38]. Other symptoms appear and other drugs are constantly taken. Verket et al. (2018) compared in their research the mean scores for SF-36 scales vitality, social functioning, and mental health between two groups of patients—with endometriosis and rheumatoid arthritis. Women with moderate to severe endometriosis scored only 33.4, 62.7, and 66.3 points whereas with rheumatoid arthritis 42.7, 68.8, and 72.6 points [39]. It is worth adding that individual conceptualization of pain in endometriosis may be affected by different social cultures in different races. Asian patients tend to normalize pain, while Caucasian patients are more likely to seek health care. The intensity of pain symptoms may be significantly lower in Chinese population than Russian or French. Moreover, Italian patients with endometriosis may suffer more from severe pelvic pain than Chinese women, suggesting a different cultural background [40,41].

Patients with endometriosis treated surgically are exposed to stress related to hospitalization and the operation itself. Approximately 88% of the patients suffer from depression and anxiety [42,43]. What is more, they often suffer from poor quality of sleep. In the observational cross-sectional study conducted by Vannuccini et al. (2018) patients who filled the ‘Patient Health Questionnaire’ (PHQ) were affected in 59% by at least one psychiatric disorder. This observation correlated with pain symptoms (*p* = 0.0026). There was no correlation between the presence of psychiatric disorders and age, BMI, parity, infertility, need for surgery, number of interventions, localization of endometriotic lesions, and systemic comorbidities. Moreover, the research also revealed that patients with severe pain showed a higher incidence of multiple psychiatric disorders (*p* = 0.026) [44]. Due to the recurrent nature of the disease, they must be frequently operated on several times during their lifetime, which increases discomfort. In the case of endometriosis of the urinary system or lower part of the gastrointestinal system, some patients undergo extensive operations leading to significant mutilation, thus greatly lowering the comfort of life. The surgical procedure within the abdominal cavity also involves the risk of postoperative adhesions formation. This problem additionally complicates the course of the main disease and is a source of more pain [45].

## 4. Sexuality and Reproductive Health

Pain also accompanies women during sexual intercourse. Approximately two thirds of women with endometriosis have sexual dysfunction [46]. Patients avoid sex, afraid of suffering [47]. Women with endometriosis desire more sexual activity than their current level. “Approximately 42.3% of endometriosis-affected women and 30.5% of the control women desired a higher frequency of sexual activity (*p* < 0.001)” [48]. The presence of adhesions, cysts, and endometriosis foci in the vagina and uterine cervix and infiltration of the surrounding tissues lead to immobilization of the reproductive organ, anatomical changes, and thus pain during penetration or the sexual act. It leads to a decrease in libido, desire, and a lack of orgasm in patients [49]. The quality of sex life is lowered. Patients are less satisfied and less relaxed after the intercourse. A decrease in sexual satisfaction is associated with a decrease in the quality of life [50]. This affects their relationship with the partner. What is more, this can even affect the stability of marriage and family. Patients suffering from endometriosis limit the number of sexual contacts, which first leads to disturbed relationships between partners. A lack of satisfaction in such an important sphere of life as sex leads to lowering of mood, anxiety disorders, and even depression [51]. A study showed that 40% of women with endometriosis-related chronic pelvic pain were unsatisfied with their overall sexual life and suffered from such symptoms as reduced frequency of sexual life, vaginal spasm, and even sexual aversion [52]. Taking into consideration that sex is one of the elements that significantly influences the comfort of human life, it becomes a great problem for women with endometriosis. Patients should understand the anatomical structure and physiological function of the female reproductive system, surgical methods, and the effect of treatment. This could reduce the unnecessary psychological burden and increase self-efficacy.

Dyspareunia is a medical term which describes recurrent or persistent pain with sexual intercourse that causes distress. It may cause a risk of sexual dysfunction, relationship distress, diminished quality of life, anxiety, and depression [53]. Dyspareunia decreases sexual activity of women with endometriosis. Some studies report that dyspareunia occurs in about 50% of women affected by endometriosis [54]. Seventy-six per cent of them demonstrate sexual distress [55]. Moreover, more than three quarters of young adult women with endometriosis and dyspareunia admit that the state negatively impacts both physical and mental health QOL (Quality of Life) scores [56]. It is related to a reduction in the number of sexual acts, deteriorates contact with the partner, lowers self-esteem, and it is also connected with infertility. The chronic fatigue syndrome, symptoms of depression, and anxiety disorders are found more often in women with endometriosis [57]. Mental symptoms appear more frequently in patients suffering from endometriosis for more than six months and in women with severe pain. It has been proven that patients with endometriosis are well educated, they obtain higher education more often, are fulfilled professionally, have a better paid job, and get married later. Additionally, the disease makes it difficult to lead an active lifestyle and it limits social contacts. Patients suffering from endometriosis are mostly women at the reproductive age, which means professionally active and usually leading a dynamic lifestyle [58]. The disease restricts their activity and personal fulfilment.

Endometriosis significantly decreases reproduction in women. It is connected with disturbances in the function of the ovaries, the egg cell itself, and the follicular fluid. In the course of the disease there is dysfunction of fimbriae of the fallopian tube and consequently disturbances in the movement of the ovum. There are problems with implantation of the fertilized egg cell (abortions). Finally, a long-lasting course leads to immobilization of the reproductive organ and anatomical changes in the small pelvis. The above-mentioned problems of biological nature are accompanied by less frequent sexual contacts, which means a decrease in fertility [59]. This is a great problem for patients, leading to lower satisfaction from sex life. Infertility leads to frustration and depressed mood. It has been proven that satisfaction from sex life in healthy women who do not have problems with conception is much higher than in patients with endometriosis, potentially infertile.

## 5. Improvement in the Quality of Life and the State of Health

Both pharmacological and surgical treatment improves the quality of life in women suffering from endometriosis [60,61,62]. In pharmacological treatment, hormonal therapy is used. It aims at a reduction of pain, such as pain during intercourse or menstrual pain. Alleviation of pain was observed in patients receiving such therapy within several months, both in patients using a combination of drugs—ethynylestradiol with dienogest, as well as dienogest alone. Improvement in well-being and the quality of sexual life was also observed in patients treated with surgery. Bastu et al. (2020) described that laparoscopic management reduced postoperative pain and increased quality of life according to pain score outcomes [63]. The effects are additionally enhanced by pharmacotherapy applied after the operation [64]. Women report a greater number of sexual contacts, a higher level of satisfaction, more orgasms, improved relationships with their partners, higher self-esteem, and a reduced number of depressive episodes. Patients cope with the disease by trying other, nonconventional methods. They use herbal therapies, TENS, psychotherapy, or physical activity. The study conducted by Boersen et al. (2021) revealed that all participants expressed strong approval of cognitive behavioural therapy, which should be added to the standard treatment [65]. Women who have accepted their disease and have learned to live with it better tolerate pain and cope with other symptoms, and they report a higher level of the quality of life, despite the presence of the disease [66]. Understanding by family members and partners may help women cope with this disease. The studies have demonstrated that the attitude of women with endometriosis has a significant influence on the course of the disease and the quality of their lives. Women with a positive attitude felt negative aspects of the disease more rarely.

## 6. Conclusions

Endometriosis affects almost every third woman at the reproductive age, especially between 25 and 29 years of age. Only 22.8% of them are asymptomatic. Some predisposing factors for endometriosis development may include low BMI, early menarche, and nulliparity. It is treated as a chronic illness, which is frequently accompanied by diseases from the autoimmune group. It has an impact on both mental and physical quality of life. The main symptom of the disease is chronic pain localized mainly in the pelvis minor [67,68,69]. The patients also complain of other burdensome symptoms. Painful defecation, regarded as a consequence of deep infiltration, is frequently pointed as positively correlated factor with the stage of the disease. Mental disturbances in the form of anxiety states and even depression with the mean age of 22.2 have been observed in women with the disease. Early screening for detection of the first signs of low mood leading to depression is highly advised. An additional problem, which leads to worsening the emotional condition of the patients, is infertility induced by the disease. It has been reported that almost every second woman with endometriosis cannot be a parent, which is indisputably one of the basic needs in human life. The phenomenon varies between 3.5% and 16.7% depending on the region of the world, women’s age, and comorbidities. It is caused not only by deterioration in the relationship with the partner and avoidance of sexual contacts due to pain, but also by the very course and complications of endometriosis at the molecular level. Based on the above observations, it is worthwhile to emphasize that women who suffer from endometriosis become pregnant later than unaffected ones, which may be associated with a higher risk of perinatal complications. Powerlessness and resigned attitude to life may also negatively affect relations with children, increasing the sense of lack of parental attention and their self-confidence. It is estimated that evolution of the disease has a negative influence on the quality of life in most of the patients. Over time and severity of the disease, women manifest more frequent fatigue in daily life, somatization, and stress.

Treatment of endometriosis, both pharmacological as well as surgical, can improve the patients’ quality of life. Pharmacotherapy significantly reduces the use of surgical procedures that deteriorate the ovarian reserve. Less invasive operative procedures are recommended. Due to 75.9% of women who report taking OTC, variety of pain sensations that do not correlate with the severity of the disease, the use of other nonconventional ways to alleviate the symptoms, such as psychotherapy, a healthy diet, physical activity, herbal therapies, and others, are recommended. The attitude of patients towards the disease and its course and acceptance of life with a chronic disease seems important. A better comprehension of all bio-psycho-social aspects implicated in women’s well-being and pain experience on endometriosis needs further research in the near future.

## Figures and Tables

**Figure 1 jcm-11-02059-f001:**
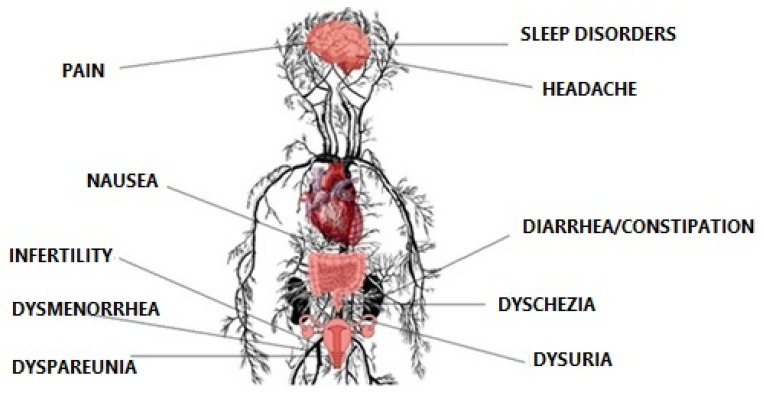
Physical symptoms of endometriosis.

**Figure 2 jcm-11-02059-f002:**
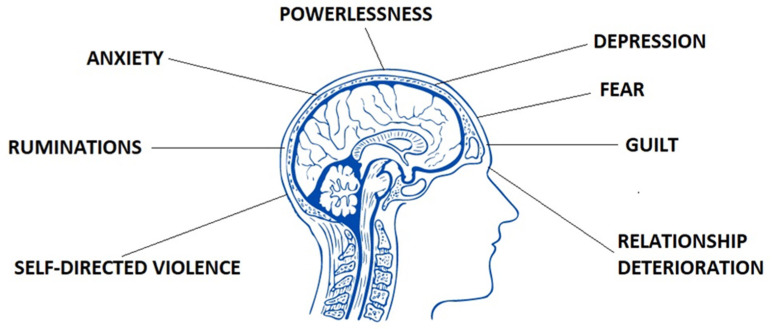
Mental states which may occur in women who suffer from endometriosis.

## Data Availability

MDPI Research Data Policies.

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
