# Peer review of "The State of Health and the Quality of Life in Women Suffering from Endometriosis"

_jcm, 2022, doi:10.3390/jcm11072059_

Round 1

Reviewer 1 Report

This review aimed to update the knowledge of the population affected with endometriosis. The authors addressed the main question and support it with relevant data.

I found this paper interesting since the approach in here is based on the actual and updated knowledge of endometriosis and how can affect the woman in this modern society. 

Line 229-230  Could be: A better comprehension of all  bio-psycho-social aspects implicated in women’s well-being and pain experience on endometriosis needs further research in the near future.

The text is clear and easy to read and the conclusions are consistent with all the evidence and arguments presented. They address the main question with introductory evidence and clinical evidence of the consequences that endometriosis have on women. 

This review can be a precursor of many other studies more on the clinical side that can study up to detail this chronic condition in order to understand not only the physical consequences but the psychological and social as well. 

Author Response

Dear Reviewer 1,

We would like to express our gratitude to the Reviewer for meaningful and helpful comments, which have made a substantial contribution to the quality of our paper.

Having studied the Reviewer’ comments and following your advice, we have decided to introduce some changes into our paper, which, we do hope, will bring some improvement to our manuscript.

"Line 229-230  Could be: A better comprehension of all bio-psycho-social aspects implicated in women’s well-being and pain experience on endometriosis needs further research in the near future."

Following your advice, we have decided to correct above sentence (line 270-272).

We are pleased that the Reviewer appreciates our work and the significance of the presented article.

Reviewer 2 Report

Peer Review

Journal of Clinical Medicine

The State of Health and the Quality of Life in Women Suffering from Endometriosis

The aim of this article is to review the current state of knowledge concerning the impact of social and medical factors on a women's population affected by endometriosis. There are topics of discussion that has been provided in this paper such as quality of life, physical and mental symptoms of endometriosis and sexuality-reproductive health. This paper offers an interesting topic regarding the correlation between women's quality of life and endometriosis. However, there are things that authors might want to consider to improve the quality of the paper.

  1. I suggest authors to discuss more about the correlation between quality of life and endometriosis in the "quality of life" section. Authors might want to provide evidences that support or perhaps contradict regarding one's quality of life affected by endometriosis. Here are some recent articles that may be useful for the authors to read and report in the "Quality of life" section

  • Muharam R, Amalia T, Pratama G, Harzif AK, Agiananda F, Maidarti M, Azyati M, Sumapraja K, Winarto H, Wiweko B, Hestiantoro A, Suarthana E, Tulandi T. Chronic Pelvic Pain in Women with Endometriosis is Associated with Psychiatric Disorder and Quality of Life Deterioration. Int J Womens Health. 2022 Feb 4;14:131-138. doi: 10.2147/IJWH.S345186. PMID: 35153516; PMCID: PMC8824289. https://pubmed.ncbi.nlm.nih.gov/35153516/

  • BieÅ„, A., RzoÅ„ca, E., Zarajczyk, M., Wilkosz, K., Wdowiak, A. and Iwanowicz-Palus, G., 2020. Quality of life in women with endometriosis: a cross-sectional survey. Quality of Life Research, 29(10), pp.2669-2677.  https://pubmed.ncbi.nlm.nih.gov/32356276/

  • Warzecha, D., Szymusik, I., Wielgos, M. and Pietrzak, B., 2020. The Impact of Endometriosis on the Quality of Life and the Incidence of Depression—A Cohort Study. International Journal of Environmental Research and Public Health, 17(10), p.3641. https://pubmed.ncbi.nlm.nih.gov/32455821/

  1. I think it would be best if authors can minimize the time span in any primary citation sentences into the last 5 years evidence-based, in order for the article to bring and discuss the current updates of the issue

  1. The conclusion need to be improved by summarizing the key findings of literatures included in this review. hoping in the near future, the readers of this article can take straight and strong message from the conclusion section

Rest of this article offers an interesting and bold recent topic that brings updated information regarding the main issue of endometriosis. Please do recheck the grammar before the article is publish. Hope this feedback can improve author's article and make it more impactful.

Author Response

Dear Reviewer 2,

We would like to express our gratitude to the Reviewer for meaningful and helpful comments which have made a substantial contribution to the quality of our paper. Having studied the Reviewer’ comments and following your advice, we have decided to introduce some changes into our paper, which, we do hope, will bring some improvement to our manuscript.

“I suggest authors to discuss more about the correlation between quality of life and endometriosis in the "quality of life" section. Authors might want to provide evidences that support or perhaps contradict regarding one's quality of life affected by endometriosis. Here are some recent articles that may be useful for the authors to read and report in the "Quality of life" section.”

Thank you very much for your comment. Regarding your suggestion, we have included recent articles in the section "quality of life" :

  1. Muraham, R.; Amalia, T.; Pratama, G.; Harzif, A.K.; Agiananda, F.; Faidarti, M.; Azyati, M.; Sumapraja, K.; Winarto, H.; Wiweko, B.; Hestiantoro, A.; Suarthana, E.; Tulandi, T.; Chronic Pelvic Pain in Women with Endometriosis is Associated with Psychiatric Disorder and Quality of Life Deterioration. Int. J Womens Health. 2022, 4,14,131-138.
  2. Bień, A.; Rzońca, E.; Zarajczyk, M.; Wilkosz, K.; Wdowiak, A.; Iwanowicz-Palus, G. Quality of life in women with endometriosis: a cross-sectional survey. Quality of Life Research. 2020,29,2669-2677.
  3. Warzecha, D.; Szymusik, I.; Wielgos, M.; Pietrzak, B. The impact of Endometriosis on the Quality of Life and the Incidence of Depression – A Cohort Study. Int. J. of Environmental Research and Public Health, 2020, 17, 3641

and basing on them we have described more correlations between women’s well-being and factors that affect their daily life.

“I think it would be best if authors can minimize the time span in any primary citation sentences into the last 5 years evidence-based, in order for the article to bring and discuss the current updates of the issue.”

Thank you very much for your comment. The references have been modified and minimized to 73 positions.

“The conclusion needs to be improved by summarizing the key findings of literatures included in this review. hoping in the near future, the readers of this article can take straight and strong message from the conclusion section”.

Thank you very much for your valuable comment. We have corrected “conclusions” section in accordance to your suggestion.

“Rest of this article offers an interesting and bold recent topic that brings updated information regarding the main issue of endometriosis. Please do recheck the grammar before the article is publish. Hope this feedback can improve author's article and make it more impactful.”

Thank you for your valuable attention. We are waiting for checking our manuscript by the MDPI English Editing Service.

We are pleased that the Reviewer appreciates the significance and topicality of the presented matter.